# Identification of Potential Mechanisms of Rk1 Combination with Rg5 in the Treatment of Type II Diabetes Mellitus by Integrating Network Pharmacology and Experimental Validation

**DOI:** 10.3390/ijms241914828

**Published:** 2023-10-02

**Authors:** Yao Liu, Jingjing Zhang, Chao An, Chen Liu, Qiwen Zhang, Hao Ding, Saijian Ma, Wenjiao Xue

**Affiliations:** Shaanxi Key Laboratory of Qinling Ecological Security, Shaanxi Institute of Microbiology, Xiying Road 76, Xi’an 710043, China; liuyao181002@163.com (Y.L.); zjj_1712@163.com (J.Z.); anchor0216@sina.com (C.A.); 13892806669@163.com (C.L.); 13772195921@163.com (Q.Z.); dhwj2010@126.com (H.D.); masaijian@163.com (S.M.)

**Keywords:** Rk1+Rg5, T2DM, network pharmacology, insulin resistance

## Abstract

In this study, we aimed to explore the potential targets and functional mechanisms of Rk1 combined with Rg5 (Rk1+Rg5) against type II diabetes mellitus (T2DM). Network pharmacology and molecular docking were used to predict and verify the targets and signaling pathways of Rk1+Rg5 against T2DM. The results were further confirmed by a db/db mouse model and a model using PA-induced L6 cells. According to network pharmacology, a total of 250 core targets of Rk1+Rg5 towards T2DM were identified; the insulin resistance signaling pathways were enriched by KEGG. Results of molecular docking indicated good binding affinity of Rk1 and Rg5 to Akt1. In vivo and in vitro studies further showed that Rk1+Rg5 is an inhibitor of skeletal muscle insulin resistance. The results showed that Rk1+Rg5 significantly improved the hyperglycemic state of db/db mice, alleviated dyslipidemia, and promoted skeletal muscle glucose uptake. This phenomenon was closely related to the alleviation of the insulin resistance in skeletal muscles. Finally, the combination activated the Akt signaling pathway and promoted GLUT4 translocation to the cell membrane for glucose uptake. Altogether, our findings, for the first time, demonstrate that the combination of Rk1 and Rg5 could be beneficial for anti-T2DM, possibly involving ameliorated insulin resistance.

## 1. Introduction

Diabetes mellitus (DM) as a chronic metabolic disease caused by raised blood glucose concentration, which is considered to be the biggest threat to human health in the 21st century [1]. Type II diabetes mellitus (T2DM) accounts for over 90% of DM and it is typically characterized by a decrease in insulin sensitivity or insulin secretion, resulting in an increase in blood glucose levels [2]. According to the World Health Organization, the number of DM patients is estimated to be 4.15 million in 2018 worldwide, and it is expected to reach 7 million by 2045 [3]. Prolonged hyperglycemia further aggravates the risk of developing a number of serious complications resulting in increased mortality [4]. Therefore, T2DM has become a major challenge in the field of non-infectious public health.

The pathogenesis of T2DM is complex and affected by various factors, but insulin resistance is one of the main pathophysiological characteristics which occurs when target organs’ exerted physiological effects become less sensitive to circulating insulin. Insulin, the important anabolic hormone in the body, can promote glucose uptake and utilization by targeting the liver, skeletal muscles, and fat [5]. Among them, skeletal muscle is an important organ for glucose metabolism and takes charge of 70–80% glucose uptake [6]. When the body absorbs energy and converts it into glucose, activation of the Akt signaling pathway further regulates the translocation of glucose transporter 4 (GLUT4) from the internal compartment to the plasma membrane of skeletal muscles [7]. Insulin may stimulate glucose transport over the plasma membrane, where approximately 85% of glucose absorption is dependent on GLUT4 [8]. Previous research indicates that the decline in ability of the transport capacity of GLUT4 may result in a reduction in glycogen synthesis, which is likely to be one of the important reasons contributing to the development of insulin resistance in skeletal muscle [9]. Therefore, the improvement of glucose metabolism in skeletal muscle can significantly accelerate glucose utility and enhance insulin sensitivity.

Maintaining a healthy weight, a balanced diet, and regulating physical exercise can effectively reduce the risk of T2DM. However, most patients need drug intervention for hypoglycemia. Thiazolidinediones and sulfonylureas, as insulin sensitizers [10], have been used for the clinical therapy of T2DM. Recently, GLP-1 receptor agonists, and DPP-IV and SGLT-2 inhibitors, as novel treatment methods to control blood glucose, have been proposed, with significant influence on the improvement of T2DM in some studies [11]. However, these drugs are usually accompanied by side effects. Specifically, long-term use of these agents could cause gastrointestinal diseases and increased cardiovascular risks [12]. Hence, finding safer hypoglycemic drugs is of great urgent need. Plant extracts are an important resource for global health care which have undeniable advantages in treating T2DM, with rich targets, significant therapeutic effects, and high safety. They have great potential in alleviating patient symptoms and blocking disease progression. Previous studies have found that baicalin attenuates insulin resistance in the skeletal muscle of T2DM mice through modulating the protein kinase B/Glycogen synthase kinase 3 beta pathway [13]. Resveratrol inhibits the PI3K/Akt signaling pathway by disrupting the interaction between IRS and their downstream proteins, thereby affecting the activity of glucose and lipid metabolism-related enzymes [14]. Blueberry intake reduces insulin resistance in the skeletal muscle and adipose tissues of obese rats [15]. Therefore, it is of considerable interest to investigate the potential anti-T2DM effect of plant extracts and the potential molecular mechanism.

Ginseng (Panax ginseng C. A. Mey), one of the most widespread natural products in the world, has been used as a Chinese medicinal herb for treating DM, and cardiovascular and inflammatory diseases, with a long history [16,17,18,19]. Ginsenosides, as the main active ingredients of ginseng, are derivatives of triterpenoid dammarane and can mainly be divided into two categories: protopanaxadiol (PPD)- and protopanaxatriol (PPT)-type saponins [20]. Our previous studies have indicated that PPD-type saponins have a hypoglycemic effect in HFD- and STZ-induced T2DM mice [21]. Ginsenoside Rk1 and Rg5 are two major active components of PPD-type saponins. However, the major anti-T2DM effect of ginsenoside Rk1 combined with Rg5 (Rk1+Rg5) still remains unclear. In this study, we aimed to explore the effective role of Rk1+Rg5 on T2DM in db/db mice. The potential molecular targets and signaling pathways were systematically elucidated via network pharmacology, transcriptome analysis, and molecular docking. Then, in vivo and in vitro assays were performed to identify their ability to treat hypoglycemia, and confirm the metabolic processes.

## 2. Results

### 2.1. Network Pharmacology Prediction

For T2DM related targets, 599, 2060, and 2360 targets were screened by OMIM, GeneCards, and DisGeNET, by the median method, respectively. After removing the duplicates, 3659 T2DM-related targets were left (Figure 1A). We intersected the screened Rk1+Rg5 active ingredient targets with T2DM disease targets, and the Venn diagram was generated by the R language. As shown in Figure 1A, Rk1+Rg5 and T2DM shared 250 common gene targets.

To explore the mechanism of Rk1+Rg5 in the amelioration of T2DM, a bubble diagram was executed, as shown in Figure 1B. The analysis showed that these signaling pathways were closely associated with T2DM, and they included the insulin resistance, cAMP, PPAR, and p53 signaling pathways. Among these signaling pathways, insulin resistance is the most significant. Next, lists of core targets were submitted to Metascape for GO enrichment analysis to clarify the possible role of candidate targets. As shown in Figure 1C, the targets between Rk1+Rg5 and T2DM were mainly involved in the response to insulin signal transduction, cellular response to insulin stimulus, protein tyrosine kinase activity, and insulin receptor signaling pathways.

As shown in Figure 1D, the target–pathway network diagram of Rk1+Rg5 and T2DM target was constructed using Cytoscape 3.7.2 software. Then, we analyzed the network topology parameters of Rk1+Rg5 in the treatment of T2DM, and the core targets were obtained. The results displayed that those top 10 proteins were involved in multiple pathways, which might be the core targets of T2DM (Figure 1E).

Next, Rk1 and Rg5 were conjugated with those 10 core targets proteins by molecular simulation software (AutoDock 4.2.6), respectively. The ligand selectivity and ligand action were analyzed using a docking simulation and molecular pathway diagram, and then the binding potential of protein–ligand was evaluated according to their comprehensive characteristics. The data of docking binding energy, GO, and KEGG pathway enrichment analyses indicate that Akt1 protein is more likely to be bound by Rk1 and Rg5 (Table 1).

Based on the analysis of molecular docking (Figure 1F), Rk1 binds with Akt1 through alkyl hydrophobicity at positions of Pro318, Leu362, Met363, Lys386, and Lys389; van der Waals forces at positions of Tyr38, Arg41, Pro42, Asp44, Glu322, Asp387, and Pro388; and hydrogen bonds at positions of Lys39, Glu40, Gln43, and Asp325. It can be seen from Figure 1G that Rg5 binds with Akt1 through alkyl hydrophobicity at positions of Tyr18, Cys310, and Lys297; van der Waals forces at positions of Leu155, Glu278, Thr312, Tyr315, Gly311, Leu295, Cys296, Arg273, Glu17, Ile84, Gly294, Lys276, Phe293, Met281, Tyr229, and Phe236; hydrocarbon bonds at positions of Leu156, Asn279, and Thr291; and hydrogen bonding at positions of Lys154, Asp274, and Glu234.

### 2.2. Effect of Rk1+Rg5 Treatment in db/db Mice

As shown in Figure 2A, a higher fasting blood glucose (FBG) level in db/db (T2DM mice) mice was observed (*p* < 0.001) compared with the db/dm group (normal mice), which illustrated that abnormal glucose metabolism was triggered in db/db mice. After 8 weeks of 50 mg/kg Rk1+Rg5 (Rk1+Rg5-L) and 100 mg/kg Rk1+Rg5 (Rk1+Rg5-H) treatment, FBG levels of db/db mice were significantly decreased by 39.24% and 55.16%, respectively. In addition, following treatment with metformin, the FBG level in the db/db group was increased by 46.16%.

And, no significant differences in body weight were observed among all groups during the 4 weeks (Figure 2B). At the end of the experiment, the body weight of the db/db group significantly increased compared to the db/dm group, whereas Rk1+Rg5 inhibited body weight gain (Figure 2C). Apparently, Rk1+Rg5-H showed better efficacy on reducing body weight compared to Rk1+Rg5-L (Figure 2C). Meanwhile, Rk1+Rg5 treatment dramatically reduced food consumption in db/db mice in a dose-dependent manner (Figure 2D).

Given the notable loss of FBG in the Rk1+Rg5 treated groups, the lipid metabolism level was also measured after treatment. As illustrated in Figure 2E–H, administration of Rk1+Rg5 and metformin significantly reduced the TG and LDL-C (*p* < 0.05), and raised the HDL-C levels compared with the db/db group (*p* < 0.05). And, evaluations of TC levels did not reveal any statistical differences between treatment and db/db groups (*p* > 0.05). The above results manifested that Rk1+Rg5 can improve lipid metabolism in db/db mice.

### 2.3. Effect of Rk1+Rg5 on the Glucose Metabolism in db/db Mice

HbA1c is an important indicator for monitoring whether the blood glucose control is up to the target. Comparing the HbA1c levels after treatment (Figure 3A), data show that the HbA1c levels of the Rk1+Rg5-L- and Rk1+Rg5-H-treated groups were decreased by 13.53% and 12.99% (*p* < 0.01), respectively, while in the metformin group, it was decreased by 0.84% (*p* < 0.05), indicating that Rk1+Rg5-treated db/db mice reduced the HbA1c levels to a greater extent than metformin intervention. What is more, the insulin levels in the db/db group were significantly lower (*p* < 0.05) than the db/dm group (Figure 3B). Following treatment with Rk1+Rg5, the serum insulin levels were increased (*p* < 0.05).

In addition, the oral glucose tolerance test (OGTT) result is shown in Figure 3C,D. After glucose administration, the blood glucose level of each group increased rapidly, reaching individual peaks at 30 min, and then gradually declined, reflecting the processes of glucose metabolism and absorption in vivo (Figure 3C). However, the response to glucose in the db/db group was weaker than that in the db/dm group, which was an apparent impairment of the glucose tolerance. Meanwhile, compared with the db/dm group, the AUC in the db/db group was significantly increased (*p* < 0.001, Figure 3D). After Rk1+Rg5-L, Rk1+Rg5-H, and metformin treatment, the AUCs were significantly reduced by 13.14%, 20.37% and 11.91%, respectively (Figure 3D).

The insulin tolerance test (ITT) was used to detect the speed and ability of the body to remove glucose after injecting the insulin to further verify the effect of Rk1+Rg5 on insulin sensitivity. As shown in Figure 3E, the ITT results showed that extrinsic insulin was less effective in db/db mice than that in db/dm mice. Supplementation of Rk1+Rg5 exhibited improved insulin tolerance; the glucose-lowering effect of insulin in the Rk1+Rg5 group was greater than that in the db/db group, and the blood glucose levels were significantly lower in the Rk1+Rg5 group at 30, 60, 90, and 120 min after insulin injection (Figure 3E). Similarly, the AUC was markedly reduced after Rk1+Rg5 treatment, compared with the db/db group (*p* < 0.01, Figure 3F). These results indicated that Rk1+Rg5 could alleviate insulin resistance in T2DM mice.

Furthermore, the insulin resistance index (HOMA-IR) and the index of homeostasis model assessment β (HOMA-β) were used to evaluate insulin resistance and β cells’ secretion function, respectively [22,23]. Rk1+Rg5-L and Rk1+Rg5-H decreased the HOMA-IR index by 45.93% and 52.85% of db/db mice and increased the HOMA-β index by 51.1% and 65.85%, respectively (Figure 3G,H, *p* < 0.05).

### 2.4. Effects of Rk1+Rg5 on Glucose Transporter in the Skeletal Muscle of db/db Mice

As the major organ of glucose metabolism, hematoxylin and eosin (H & E) staining was conducted on db/db mice to observe the morphology of skeletal muscle tissue in each group, and to evaluate the effect of Rk1+Rg5 on the morphology of skeletal muscle in db/db mice. As shown in Figure 4A, compared to the db/dm group, there is a disorganized arrangement of muscle fibers, blurred bundle membrane boundary, and smaller diameter of muscle fibers, showing that they were disordered. After Rk1+Rg5 treatment, the damage to muscle fibers was significantly improved, with a neat arrangement and a clear fascicular structure, compared to the db/db group. Similarly, we also observed that the islets were round or oval, evenly distributed, regular in shape, and had a large number of islet cells in the db/dm group. The db/db group showed that the cell volumes were smaller and distributed diffusely. Rk1+Rg5 treatment alleviated the above phenomena (Appendix A).

Muscle glycogen is the main form of glucose storage in skeletal muscle, and periodic acid-Schiff (PAS) staining can stain glycogen purple/red. As shown in Figure 4B, compared with the db/dm group, the glycogen content of the db/db group of mice was significantly reduced (*p* < 0.05), while Rk1+Rg5 treatment significantly increased the glycogen content of db/db mice (*p* < 0.01).

The glucose absorption by skeletal muscles is mainly achieved through GLUT4 protein transport. Western blotting analysis confirmed that Rk1+Rg5 significantly increased the expression of GLUT4 (Figure 4C). Meanwhile, the expression of GLUT4 protein in the skeletal muscle of db/db mice was further detected by immunofluorescence staining. As shown in Figure 4D, compared with the db/dm group mice, the expression of GLUT4 in the db/db group mice was significantly reduced. After Rk1+Rg5 treatment, the expression of GLUT4 protein in the skeletal muscles of db/db mice significantly increased, and more translocation to the cell membrane was also observed in the Rk1+Rg5 treatment group (Figure 4D). These results suggested that Rk1+Rg5 may treat T2DM by regulating the glucose transporter in the skeletal muscle of db/db mice.

### 2.5. Effects of Rk1+Rg5 on Glucose Uptake in L6 Cells

Figure 5A revealing the process of L6 cells for differentiation of skeletal muscle cells, using α-Sarcometic actin to identify that the L6 cells have successfully differentiated into skeletal muscle cells. As shown in Figure 5A, α-Sarconic actin was rarely expressed in L6 cells in normal medium. After 7 days of replacement with differentiation medium, α-Sarcometic actin protein is highly expressed. Those results indicate that L6 cells successfully differentiated into skeletal muscle cells.

Next, we determined the appropriate concentration of palmitic acid (PA) by cytotoxicity assay. When the concentration of PA exceeds 0.8 mM, the activity of L6 cells significantly decreases (Figure 5B, *p* < 0.05), indicating that the maximum safe dose of PA on L6 cells is 0.8 mM. Thus, the concentrations of 0.4, 0.6, and 0.8 mM PA were used in subsequent glucose uptake assays.

In order to determine the optimal induction time of PA, L6 cells were induced with serum-free DMEM containing different concentrations of PA (0.4, 0.6, 0.8 mM) for 12, 24, and 48 h, respectively. As shown in Figure 5C, after 24 h of induction with 0.8 mM PA, the glucose consumption of L6 cells was significantly reduced (*p* < 0.05). Therefore, the optimal PA-induced concentration of insulin resistance in the cell model (IR-L6) was 0.8 mM, and the optimal induced time was 24 h.

In addition, we found that the maximum safe doses of Rk1+Rg5 on IR-L6 cells were 0.03 and 0.06 mM, by MTT experiments (Figure 5D). Afterwards, we investigated if Rk1+Rg5 was able to promote the glucose uptake in the IR-L6 cell model. As shown in Figure 5E, compared with the con group, the glucose consumption of IR-L6 cells was significantly reduced, while after Rk1+Rg5 treatment, the glucose consumption of IR-L6 cells was significantly increased (*p* < 0.01). Then, we further explored the impacts of Rk1+Rg5 on the glycogenesis in the IR-L6 cell model. As shown in Figure 5F, the glycogen content was markedly higher in the Rk1+Rg5 group compared with the PA-induced group (*p* < 0.01). In addition, insulin signaling pathway-related protein p-Akt was up-regulated considerably in the Rk1+Rg5 treatment group compared to the PA-induced group (Figure 5G). These results indicated that Rk1+Rg5 has a strong inhibitory effect on the glucose uptake of IR-L6 cells and its molecular mechanisms are possible due to the role of Akt signaling pathway activation.

### 2.6. Effect of Rk1+Rg5 on the Potential Mechanism in T2DM

In order to elucidate the potential mechanism of Rk1+Rg5-mediated T2DM, transcriptome analysis was performed in PA-induced L6 cells. As shown in Appendix A, compared with PA group, Rk1+Rg5 treatment significantly changes 134 genes. Among them, 105 genes were significantly upregulated and 29 genes were significantly downregulated. The differential expression clustering analysis heat map showed that Rk1+Rg5 treatment reversed the differences of gene expression in the PA group compared to the con group (Figure 6A). Next, GO and KEGG enrichment analyses were conducted on differentially expressed genes to investigate the mechanism of Rk1+Rg5. KEGG analysis showed that Rk1+Rg5 mainly regulated the PI3K-Akt signaling pathway, FoxO signaling pathway, MAPK signaling pathway, AMPK signaling pathway, insulin signaling pathway, and insulin resistance (Figure 6B). GO analysis showed that Rk1+Rg5 was mainly involved in the biological processes of protein transport, intracellular protein transport, and protein stabilization (Figure 6C). The main molecular functions include molecular function, protein binding, identical protein binding, ATP binding, RNA binding, and protein domain specific binding (Figure 6C). Next, we focused on differentially expressed genes using key driver genes’ (Insr, Irs1, Srebf2, PI3K, Akt, Glut4) analysis to further investigate the potential mechanism of Rk1+Rg5 in T2DM (Figure 6D); these genes were related to immune responses of the insulin signaling pathway, glucose transporters, and insulin resistance. Therefore, Rk1+Rg5 attenuates T2DM symptoms primarily by regulating glucose metabolism and insulin resistance.

## 3. Discussion

T2DM, as a persistent metabolic disorder, is characterized by insulin resistance. Moreover, lipid accumulation and abnormal glucose metabolism promote the occurrence and development of T2DM. This study found that insulin resistance regulates insulin signal transduction by mediating PI3K/Akt signaling pathways and promoting glucose transport. Therefore, the amelioration of insulin resistance is a key pathway to control T2DM. Rk1 and Rg5 are rare ginsenosides, which are obtained by the deglycosylation of protopanaxadiol-type saponins. Our previous studies found that panaxadiol saponins can improve T2DM [21]. However, the direct absorption and utilization of panaxadiol saponins with more glycosyl are difficult, causing a great reduction in their biological activity [24]. The rare ginsenosides Rk1 and Rg5, obtained after transformation, show higher pharmacological activity due to their reduced glycosyl number, increased hydrophobicity, and enhanced cellular penetration. Therefore, we explored the hypoglycemic effect and mechanism of Rk1+Rg5 combination therapy for T2DM in db/db mice. Here, we found that Rk1 combined with Rg5 reversed the hyperglycemia symptoms in db/db mice.

Network pharmacology was first used to predict the target of Rk1 and Rg5 in the treatment of T2DM. The interaction between Rk1+Rg5, target, and disease was investigated by building the network and the analysis of GO gene classification annotation, as well as KEGG pathway enrichment. GO and KEGG enrichment analyses showed that Rk1+Rg5 was mainly enriched in the insulin resistance, cAMP, PPAR, and p53 signaling pathways. Metabolic pathways mainly focused on insulin signal transduction, the cellular response to insulin stimulus, protein tyrosine kinase activity, and the insulin receptor signaling pathway. These pathways are closely related to the occurrence of insulin resistance, which is the main pathogenic factor of T2DM, manifesting in a decreasing insulin sensitivity of target tissues (liver, skeletal muscle, and fat), resulting in a decreased ability for glucose absorption [25]. The cAMP signaling pathway is an important pathway to stimulate insulin secretion, which can promote the absorption of calcium, causing insulin release from β cells [26].

Based on the results of network pharmacology, the targets of Rk1 combined with Rg5 for treating T2DM were enriched in the insulin resistance pathway. Therefore, we next aimed to verify the Rk1 combination with Rg5 to improve insulin resistance by using the db/db mouse model, so as to clarify the mechanism of improving insulin resistance. First, the effects of Rk1+Rg5 on FBG and HbA1c were tested in db/db mice. Elevation of FBG and HbA1c are important symptoms of T2DM. HbA1c can reflect the long-term blood glucose level of the body. OGTT can simulate postprandial hyperglycemia, which is an effective method to evaluate glucose tolerance [27]. In this study, after Rk1+Rg5 intervention, the levels of FBG and HbA1c decreased to varying degrees, and the glucose tolerance increased, indicating that Rk1+Rg5 can improve the glucose metabolism disorder in db/db mice. Insulin resistance is the typical characteristic of T2DM and throughout the whole process of T2DM. Intervention of insulin resistance is one of the important measures to prevent and treat T2DM [28]. At present, HOMA-IR has become a common measure to evaluate insulin resistance. Increased fasting insulin levels due to a reduction in insulin receptors leads to insulin resistance, which manifests as increased HOMA-IR [29]. In this study, after Rk1+Rg5 intervention, the HOMA-IR levels significantly decreased, and the serum insulin levels notably increased in db/db mice. ITT can further evaluate insulin sensitivity [30]. Our data suggested that Rk1+Rg5 enhanced insulin sensitivity in db/db mice. The above results indicated that Rk1+Rg5 not only regulated the glucose homeostasis, but also improved insulin resistance in db/db mice.

Among the main target tissues of insulin resistance, skeletal muscles are responsible for ingesting and metabolizing over 80% of glucose, making it an important tissue for regulating glucose homeostasis [31]. Insulin stimulates the insulin receptor on the skeletal muscle cells, which subsequently activates the PI3K/Akt signaling pathway to mediate GLUT4 membrane translocation [32]. Reduction in GLUT4 was the direct cause of insulin resistance [33]. In our study, we noticed that Rk1+Rg5 significantly increased glycogen content, as well as leading to a rise in GLUT4 expression and membrane translocation in the skeletal muscle of db/db mice. Moreover, we observed that Rk1+Rg5 significantly promoted glucose uptake in PA-induced L6 cells. These results indicated that Rk1+Rg5 improves insulin resistance in skeletal muscles by promoting glucose transport. Further transcriptome analysis showed that Rk1+Rg5 mainly regulates the Akt signaling pathway and insulin resistance. In general, insulin increases GLUT4 expression by regulating the Akt signaling pathway, thereby accelerating glucose uptake and promoting its metabolism in skeletal muscle. These signaling mechanisms are disrupted when skeletal muscle insulin resistance occurs, leading to a decrease in the insulin sensitivity and glucose absorption [34], which caused further damage to pancreatic β cells and impaired glucose tolerance, driving the development of T2DM. Akt1, located at a critical position in the PPI network diagram, also participates in the key Akt signaling pathway. Akt1 is a serine/threonine protein kinase with an improvement effect on β cell function and glucose homeostasis [35], and its deficiency can increase insulin sensitivity. Considering that Rk1+Rg5 may directly target the Akt signaling pathway, we analyzed the binding mechanism of Rk1+Rg5 and Akt1 using molecular docking. Those results further verified that the key mechanism underlying the anti-T2DM effect of Rk1+Rg5 may be improving insulin resistance in skeletal muscle through the regulation of Akt1.

Rk1 and Rg5 are the major compounds of protopanaxadiol-type saponins [36]. Network pharmacology analysis revealed that Rk1 and Rg5 contributed the most to the anti-T2DM effects via the insulin resistance pathway. Rk1 and Rg5 occur as isomers pairs in ginsenosides. Due to the difficulty in isomer separation, Rk1 and Rg5 were given to db/db mice as a mixture for in vivo and in vitro evaluation. In db/db mice, Rk1+Rg5 regulated the levels of FBG and lipids, and improved glucose metabolism and insulin resistance by modulating the Akt signaling pathway, showing similar effects to Rg5 (Appendix A). The results demonstrated the key contribution of the Rk1+Rg5 combination on the therapy for T2DM.

## 4. Materials and Methods

### 4.1. Materials and Chemicals

Ginsenosides Rk1 and Rg5 (purity > 95%) were derived from PuRuiFa technology (Chengdu, China). Figure 7A shows the chemical structure of Rk1 and Rg5, respectively. Glucose test strips and glucometer were purchased from Roche (Mannheim, Germany). The HbA1c and insulin ELISA kit were purchased from Shanghai Enzyme linked Biotechnology Co., Ltd. (Shanghai, China). We purchased metformin from Sigma-Aldrich. GLUT4 (66846-1-Ig), Akt (60203-2-Ig), p-Akt (80455-1-RR), and GAPDH (10494-1-AP) were purchased from Proteintech Co., Ltd. (Wuhan, China).

### 4.2. Potential Target Proteins, PPI, GO and KEGG Pathway Enrichment Analysis

To export the information of drug–target, the SwissTarget Prediction (http://www.swisstargetprediction.ch/), accessed on 21 February 2022, PubMed (http://pubmed.cn/), accessed on 21 February 2022, UniProt (https://www.uniprot.org/), GeneCards (https://www.genecards.org), accessed on 21 February 2022, OMIM (https://omim.org/), accessed on 21 February 2022, and DisGeNET (https://www.disgenet.org/) accessed on 21 February 2022, databases were used to find seed genes, respectively. After combining, the redundancy was removed to reach related targets of T2DM.

In order to clarify the interaction of drug-related target with the target of T2DM, we used the R language to draw the Venn diagram. Then, we submitted the intersection targets to the STRING11.0 database (https://string-db.org) to construct the PPI network model [37]. The interaction network of Rk1 and Rg5 in T2DM was constructed by using Cytoscape 3.7.2. Meanwhile, GO and KEGG pathway enrichment analysis was conducted with the Metascape database (http://metascape.org/gp/index.html) [38,39]. Pathways with *p* value ≤ 0.01 according to KEGG biological pathway enrichment analysis were selected, and then the top 10 pathways with count values were further extracted.

### 4.3. Molecular Docking

The binding site between Rk1/Rg5 and Akt1 was identified using AutoDock 4.2.6 software. The 3D structure of Rk1 or Rg5 was drawn in Chem 3D Ultra 14.0, and the energy of small molecules was minimized using the MM2. The proper structure of Akt1 was selected from the Protein Data Bank (PDB ID: 7NH5). Molecular docking was used to evaluate the docking pose [40].

### 4.4. Animals and Experimental Design

Eight-week-old male Lepr^db^ mutant db/db mice (C57BL/KsJ background, Shanghai Model Organisms Center) and nondiabetic male littermates (db/dm) were purchased from the Shanghai Research Center for Model Organisms (Shanghai, China). All of the procedures performed on animals in the present study were in accordance with Chinese law. The animal experiments were approved by the Animal Ethics Committee of Northwest University (Xi’an, Shaanxi, China, approval no. NWU20220315). Body weight and food intake were recorded every day. After 1 week of adaptation, the mice were randomly assigned into 5 groups with 10 mice in each: (1) normal mice (db/dm group), (2) db/db mice (T2DM group), (3) db/db + 50 mg/kg Rk1+Rg5 (Rk1+Rg5-L), (4) db/db + 100 mg/kg Rk1+Rg5 (Rk1+Rg5-H), and (5) db/db + 300 mg/kg metformin (met). An experimental process of the animal study is shown in Figure 7B. Fresh blood samples were collected directly from the orbital sinus, and the serum was isolated and immediately stored at −80 °C for subsequent analysis. Tissue samples from the pancreas and skeletal muscle were harvested for histological analysis.

### 4.5. Analysis of FBG, OGTT, and ITT

FBG levels of mice were monitored by glucometer weekly. One day before the animals were sacrificed, glucose measurements from the OGTT and ITT were undertaken [41]. Briefly, the glucose level is deployed as the basal blood glucose level (0 min) after 6 h fasting. After the gavage of glucose (2 g/kg) or intraperitoneal injection with insulin (0.8 U/kg), the levels of blood glucose were measured at 30, 60, 90, and 120 min, respectively. The AUCs were calculated based on glucose levels at different times.

### 4.6. Biochemical Analyses

After 8 weeks of treatment, the mice were executed after 12 h of fasting and anesthetized with sodium barbiturates. Serum was routinely isolated for further tests. Commercial ELISA kits (Mlbio, Shanghai, China) were used to measure mouse insulin and HbA1c levels according to the manufacturer’s instructions. Serum levels of TC, TG, HDL-C, and LDL-C were detected according to the manufacturer’s instructions (Nanjing Jiancheng Biology Engineering Institute, Nanjing, China).

### 4.7. HOMA-IR and Index of HOMA-β Analysis

HOMA-IR is a mathematical model, which can be used to evaluate the insulin resistance index. HOMA-β is used to evaluate cells’ secretion function.
HOMA-IR=FBG (mM) × fasting insulin (mIU/L)/22.5
HOMA-β=20 × fasting insulin (mIU/L)/[FBG (mM)−3.5]

### 4.8. Histopathological Analysis

Skeletal muscle and pancreas tissues were stored in formalin-fixed paraffin embedded, then prepared into 3–5 μm thick sections. Tissue sections were stained for histochemistry using H & E and PAS staining. Direct histopathological observation was conducted under a TE 2000 fluorescence microscope (Nikon, Japan).

### 4.9. Immunofluorescence of Skeletal Muscle

The sections were incubated in 3% H_2_O_2_ solution at room temperature for 15 min, followed by 5–10% sealing solution for 30 min. The blocking solution was washed off, primary antibodies were added and the sections were treated for 4 h. Then, secondary antibodies were incubated for 1 h. The nuclei were stained with DAPI. Fluorescence signals were captured using laser scanning confocal microscopy (Leica, Wetzlar, Germany).

### 4.10. Cell Culture and Differentiation

The rat skeletal muscle myoblast cell line (L6) was purchased from the American Type Culture Collection (ATCC, Manassas, VA, USA). L6 cells were cultured in DMEM culture medium containing 10% FBS and 1% penicillin-streptomycin and incubated in 37 °C, 5% CO_2_, and passaged when the cells fused to 70–80%. When the cells reached 80% confluence, the culture medium was replaced with DMEM containing 2% fetal bovine serum to initiate myogenic differentiation. The culture medium was updated every 2 days for the first 4 days, and every day for the next 3 days. Mature myotubes were used for subsequent experiments.

### 4.11. PA-Induced Insulin Resistance Model in L6 Cells

L6 cells were seeded in 96-well plates at 7 × 10^3^ cells/well after 7 days of differentiation. Then, serum-free DMEM containing different concentrations of PA (0, 0.2, 0.4, 0.6, 0.8, 1.0 mM) was added to the cells for 12, 24, and 36 h. After removal of medium, cells were incubated with 5 ug/mL MTT for 3 h. Next, the supernatant was discarded, 150 μL DMSO was added, and OD values were detected at 490 nm using a microplate reader. Finally, the cell survival rate was calculated, and the maximum safe dose of PA was determined.

### 4.12. Glucose Consumption

Cells (1 × 10^4^ cells per well) were seeded into 24-well plates and experiments were conducted after differentiation. The experiment was divided into the normal group (con), model group (PA), Rk1+Rg5 low-dose group (0.03 mM), and Rk1+Rg5 high-dose group (0.06 mM). The con group was cultured with DMEM medium, the PA group was induced with 0.8 mM PA, while treatment groups were supplemented with different concentrations of Rk1+Rg5 (0.03 mM and 0.06 mM). After induction for 24 h, a glucose kit was used to detect the glucose content in the supernatant of each well. The difference in glucose content between the sample group and the con group is the glucose consumption.

### 4.13. Western Blotting Analysis

After treatment of Rk1+Rg5, total proteins were extracted with RIPA buffer. A BCA assay kit was used to test the protein concentration (Solarbio Science & Technology, Beijing, China). Protein samples were separated by SDS-PAGE and then transferred onto polyvinylidene fluoride membranes. After blocking with 5% nonfat milk, GLUT4 (1:1000 dilution), Akt (1:5000 dilution), and GAPDH (1:10,000 dilution) were incubated overnight at 4 °C. Then, appropriate secondary antibodies were incubated for 1 h. Western blots were analyzed by an ECL system (PerkinElmer, Waltham, MA, USA)

### 4.14. RNA Sequence Analysis

After treatment, total RNA of L6 cells was extracted, and RNA sequencing was performed. An RNA purification kit was used to remove potential contamination from the sample. Differentially expressed genes were screened with DESeq2 software; genes with *p* < 0.05 and |log2 fold change| ≥ 1 were used as differentially expressed genes. GO function and KEGG pathway analyses of target genes were performed using the TopGO software and KEGG database, respectively.

### 4.15. Data Processing and Statistical Analyses

The experimental data are presented as means ± SEM. The statistical analysis was performed using SPSS 17.0 (SPSS, Chicago, IL, USA). A one-way analysis of variance (ANOVA) test was performed for multiple comparisons using GraphPad Prism 9.0. (Graphpad, La Jolla, CA, USA). In this study, all values *p* < 0.05 were considered statistically significant.

## 5. Conclusions

Based on network pharmacology and molecular docking, this study revealed the key targets of Rk1+Rg5 intervention in T2DM, as well as the main signaling pathways (the insulin resistance signaling pathway, cAMP signaling pathway, PPAR signaling pathway, and p53 signaling pathway). In addition, we demonstrated the potential therapeutic effect of Rk1+Rg5 in db/db mice through in vivo and in vitro experiments and explored its mechanism by focusing on the insulin resistance pathway. Administration of Rk1+Rg5 could reduce FBG and lipid levels through partially regulating glucose metabolism and improving the insulin resistance of skeletal muscles. This work provided a theoretical basis for further mechanism studies on the anti-T2DM effects of Rk1+Rg5.

## Figures and Tables

**Figure 1 ijms-24-14828-f001:**
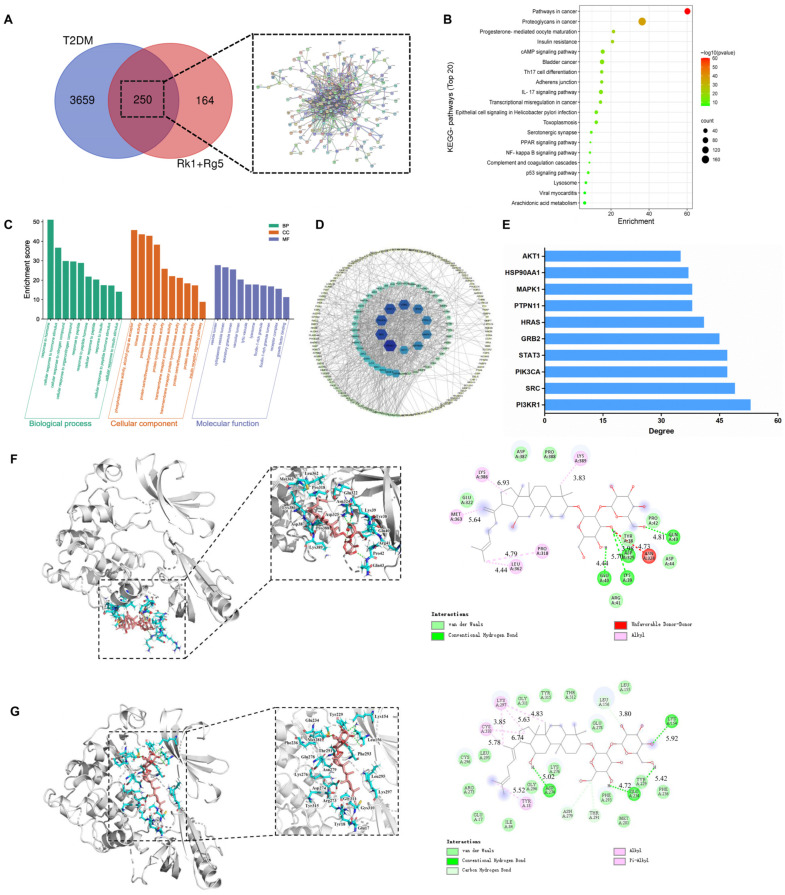
Network pharmacology analysis of Rk1+Rg5 on T2DM. (**A**) The screening process of core targets of Rk1+Rg5 acting on key nodes of T2DM. (**B**) Bubble chart of KEGG pathways’ analysis. (**C**) GO enrichment analysis. (**D**) Construction of key target–pathway network. (**E**) The proportion of key targets in the top 10 pathways. (**F**) Binding mode of Rk1 with Akt1 (7NH5) by molecular models. (**G**) Binding mode of Rg5 with Akt1 (7NH5) by molecular models.

**Figure 2 ijms-24-14828-f002:**
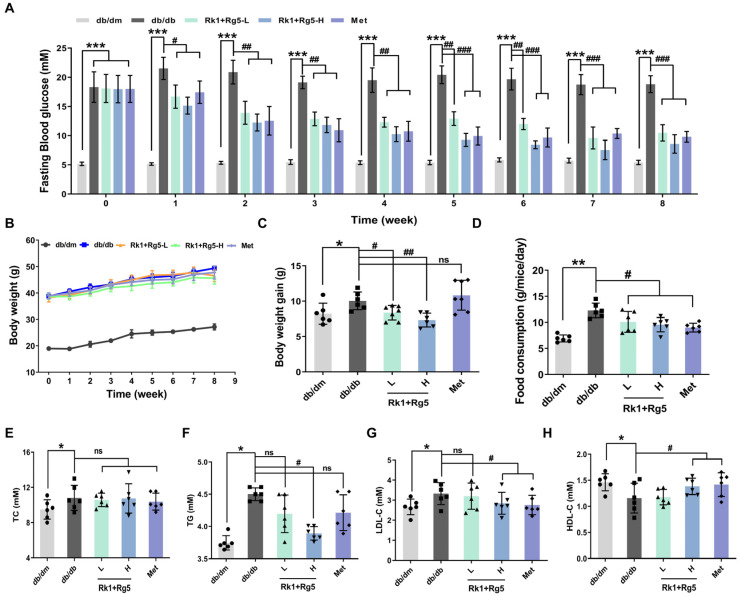
Rk1+Rg5 alleviates the symptoms of hyperglycemia in db/db mice. (**A**) FBG levels during experiments. (**B**) Change in body weight during experiments. (**C**) Body weight gain. (**D**) Food consumption. (**E**–**H**) lipid levels (TC, TG, LDL-C, HDL-C). Data are shown as mean ± SEM. *** *p* < 0.001, ** *p* < 0.01, * *p* < 0.05 compared with the db/dm group. ### *p* < 0.001, ## *p* < 0.01, # *p* < 0.05 compared with the db/db group. ns: not significant (*p* > 0.05).

**Figure 3 ijms-24-14828-f003:**
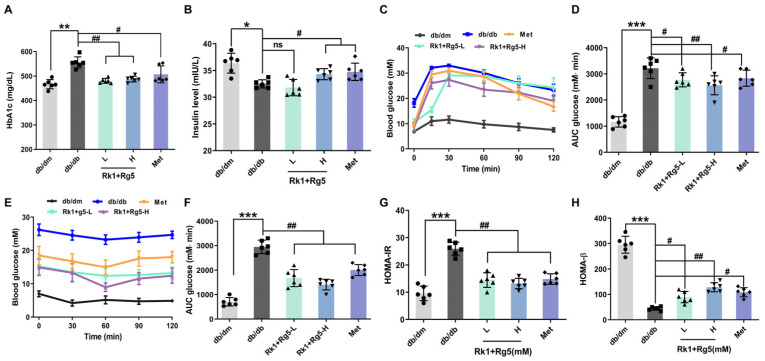
Effects of Rk1+Rg5 on the serum levels. (**A**) HbA1c level. (**B**) Fasting insulin. (**C**) OGTT. (**D**) AUC. (**E**) ITT. (**F**) AUC. (**G**) HOMA-IR. (**H**) HOMA-β. db/db mice were treated with Rk1+Rg5 (50, 100 mg/kg), or metformin (300 mg/kg) for 8 weeks. Data are shown as mean ± SEM. *** *p* < 0.001, ** *p* < 0.01, * *p* < 0.05 compared with db/dm group. ## *p* < 0.01, # *p* < 0.05 compared with the db/db group.

**Figure 4 ijms-24-14828-f004:**
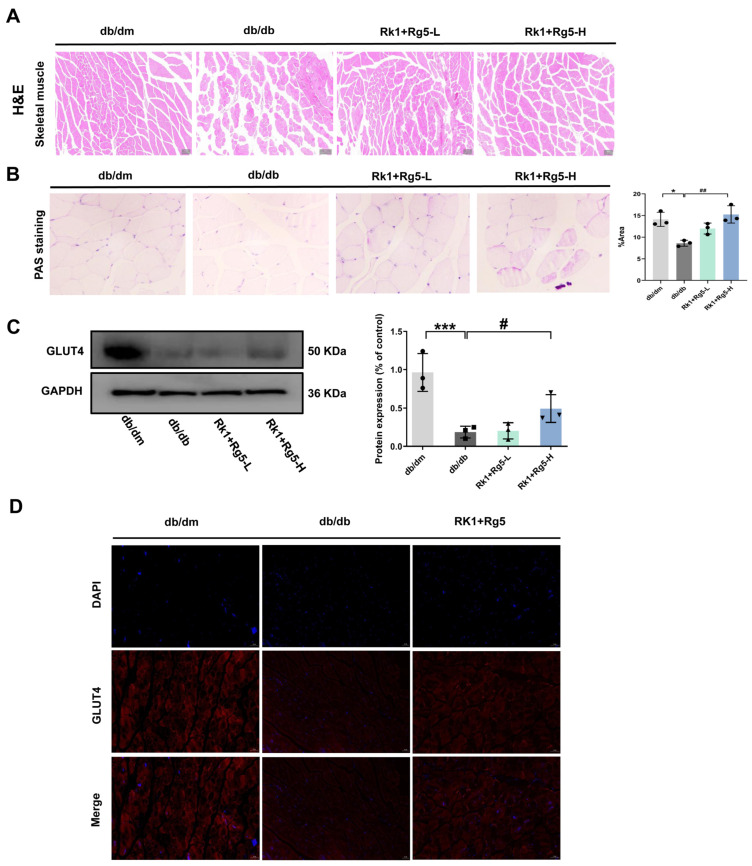
Effect of Rk1+Rg5 on glucose metabolism in skeletal muscle of db/db mice. (**A**) H & E staining of skeletal muscle. (**B**) PAS staining of skeletal muscle. (**C**) Expression of GLUT4 in skeletal muscle by Western blotting and quantitative results. (**D**) Immunofluorescence staining of GLUT4 in skeletal muscle. Data are shown as mean ± SEM (*n* = 3). *** *p* < 0.001, * *p* < 0.05 compared with the db/dm group. ## *p* < 0.01, # *p* < 0.05 compared with the db/db group.

**Figure 5 ijms-24-14828-f005:**
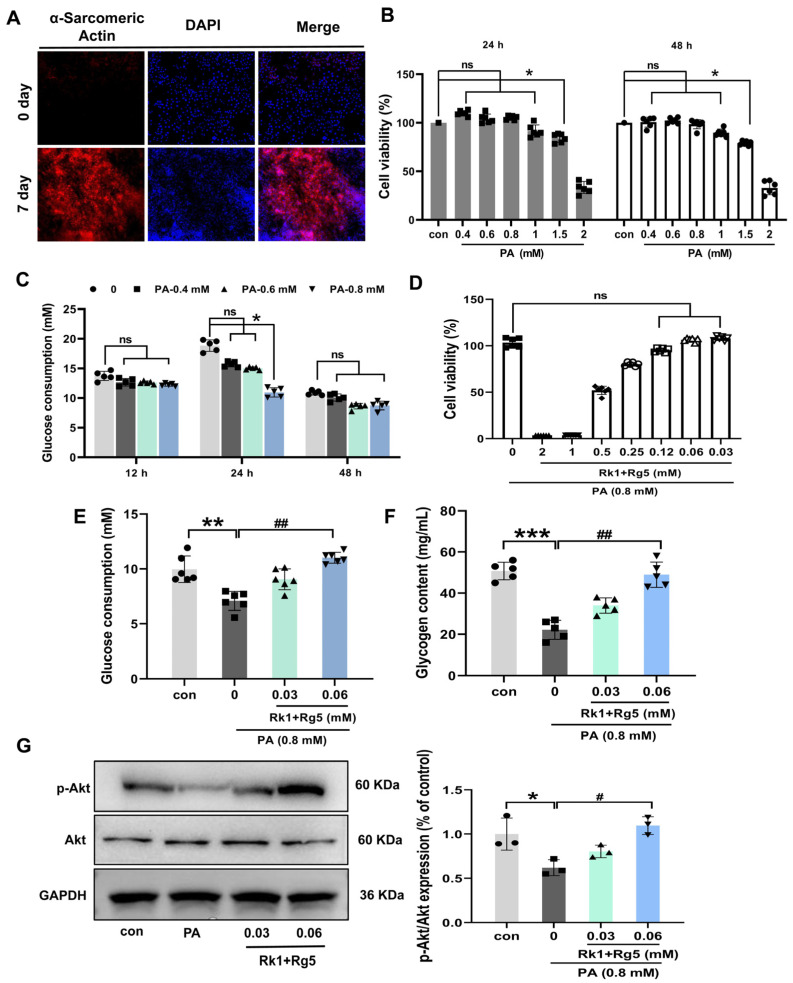
Effects of Rk1+Rg5 on skeletal muscle glucose uptake in PA-induced L6 cells. (**A**) Detection of α-sarcomeric actin protein expression in L6 cells by immunofluorescence staining. (**B**) Cell viability for PA in L6 cells. (**C**) Screening for optimal induction conditions of PA by glucose consumption. (**D**) Cell activity for Rk1+Rg5 in PA-induced L6 cells. (**E**) Glucose consumption for Rk1+Rg5 in IR-L6 cells. (**F**) Glycogen content for Rk1+Rg5 in IR-L6 cells. (**G**) Expression of p-Akt in PA-induced L6 cells was analyzed by Western blotting and quantitative results are shown. Data are shown as mean ± SEM (*n* = 3). *** *p* < 0.001, ** *p* < 0.01, * *p* < 0.05 compared with the con group. ## *p* < 0.01, # *p* < 0.05 compared with the PA group. ns: not significant (*p* > 0.05).

**Figure 6 ijms-24-14828-f006:**
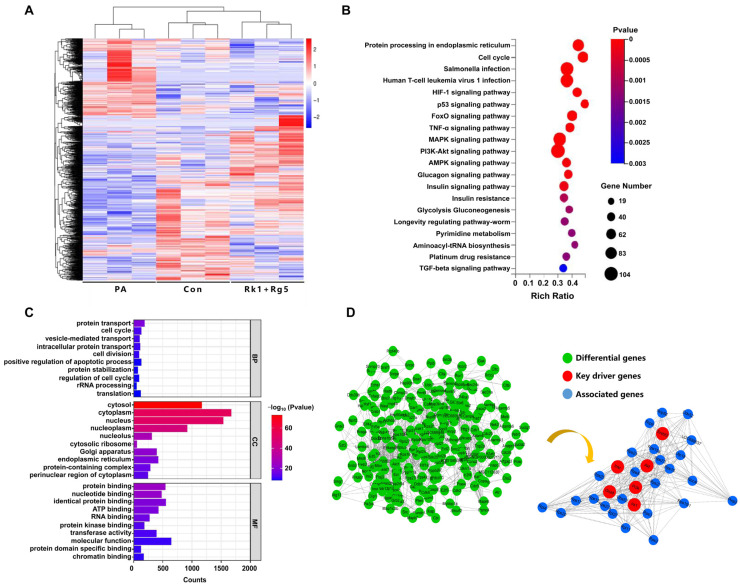
Effects of Rk1+Rg5 transcriptome alterations in PA-induced L6 cells. (**A**) Heatmap analysis of differential gene expression in con group, PA, and Rk1+Rg5 group (blue and red indicate expression, down- and up-regulation, respectively). (**B**) KEGG enrichment method to analyze the potential anti-T2DM pathways of Rk1+Rg5. (**C**) GO enrichment to investigate the possible biological process. |log2FC| ≥ 1, *p*-value ≤ 0.05. (**D**) PPI network analysis of driver genes.

**Figure 7 ijms-24-14828-f007:**
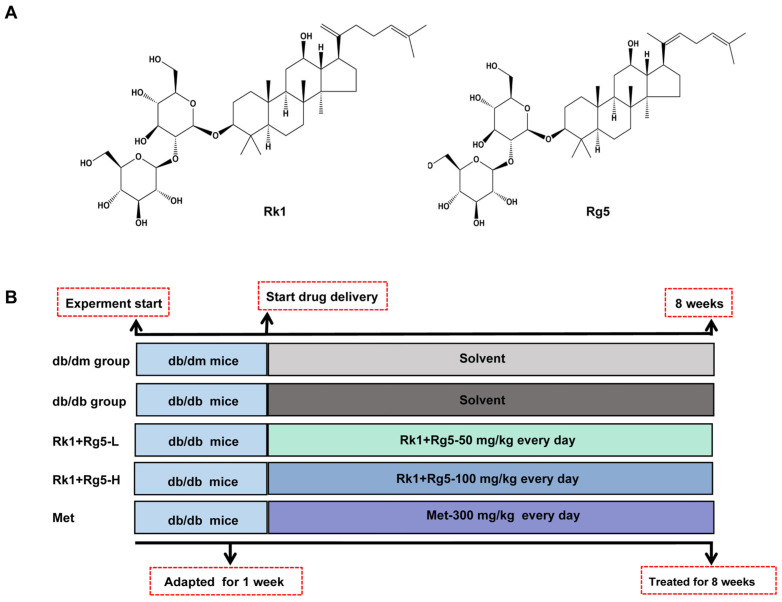
(**A**) The chemical structure of ginsenoside Rk1 and ginsenoside Rg5. (**B**) Schematic diagram of the design of db/db mice experiment.

**Table 1 ijms-24-14828-t001:** Docking studies of Rk1 or Rg5 and key targets.

Receptor	Binding Interaction Energy (kcal/mol)		VDW Interaction Energy (kcal/mol)		Electrostatic Interaction Energy (kcal/mol)	
	Rk1	Rg5	Rk1	Rg5	Rk1	Rg5
PI3KR1	−1.64	−1.06	−6.89	PI3KR1	−1.64	−1.06
SRC	−4.24	−5.06	−9.12	SRC	−4.24	−5.06
PIK3CA	−4.15	−3.65	−9.28	PIK3CA	−4.15	−3.65
STAT3	−0.99	−1.71	−6.29	STAT3	−0.99	−1.71
GRB2	−5.62	−5.81	−10.71	GRB2	−5.62	−5.81
HRAS	−5.13	−6.19	−10.11	HRAS	−5.13	−6.19
PTPN11	−4.31	−5.14	−9.6	PTPN11	−4.31	−5.14
MAPK1	−5.42	−4.52	−10.49	MAPK1	−5.42	−4.52
HSP90AA1	−4.86	−2.84	−9.96	HSP90AA1	−4.86	−2.84
Akt1	−4.95	−5.46	−9.68	Akt1	−4.95	−5.46

## Data Availability

The data presented in this study are available on request from the corresponding author.

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
