# Peer review of "Identification of Potential Mechanisms of Rk1 Combination with Rg5 in the Treatment of Type II Diabetes Mellitus by Integrating Network Pharmacology and Experimental Validation"

_ijms, 2023, doi:10.3390/ijms241914828_

Round 1

Reviewer 1 Report

minor revision

1. Modification of abstract

The abstract primarily focuses on network analysis methodology. It is advisable to incorporate result-oriented descriptions to enhance the content base on experimental results.

2. Choice of Control in Diabetes Experiments:

The abbreviations "db/db mice" and "db/dm" lack clarity without additional context. It is unclear what these abbreviations or terms signify. To provide accurate descriptions, more specific information or definitions for these terms is needed.

2. Clarification on Figure 4:

In Figure 4:  High-magnification images must be added to explain the appearance of the atrophy of nucleus (yellow arrows). And it would be preferable if photographs of all the groups were provided.

3. Key Driver Genes in Results 2.6:

In the section labeled Results 2.6, it would be helpful if you could specify which genes you consider as the 'key driver genes'. This will provide a clearer understanding of the molecular pathways being targeted.

4. Discrepancy in PI3K/AKT Signaling:

The abstract mentions that "the possible mechanism could be attributed to the Rk1+Rg5 regulated insulin resistance by modulating PI3K/Akt signal pathway." However, I did not find specific experimental evidence either in vitro or in vivo related to the PI3K/AKT signaling pathway. Given the importance of verification in network pharmacology, it would be beneficial if you could address this.

In conclusion, I believe that the study presents interesting findings on the potential mechanisms of Rk1 combined with Rg5 in the treatment of T2DM. Addressing the above queries and concerns will further strengthen the manuscript's contribution to the field. 

Author Response

Response to Reviewer #1:

  1. Modification of abstract

The abstract primarily focuses on network analysis methodology. It is advisable to incorporate result-oriented descriptions to enhance the content base on experimental results.

Response: Thank you for your valuable suggestions. Done as suggested. We have revised the abstract in the revised manuscript in red.

  1. Choice of Control in Diabetes Experiments:

The abbreviations "db/db mice" and "db/dm" lack clarity without additional context. It is unclear what these abbreviations or terms signify. To provide accurate descriptions, more specific information or definitions for these terms is needed.

Response: Thank you for your valuable suggestions. We have revised the definitions of these terms in the revised manuscript in red.

  1. Clarification on Figure 4:

In Figure 4: High-magnification images must be added to explain the appearance of the atrophy of nucleus (yellow arrows). And it would be preferable if photographs of all the groups were provided.

Response: Thank you for your valuable suggestions. Done as suggested. We have added the high-magnification images in supporting information of figure S1. 
3. Key Driver Genes in Results 2.6:

In the section labeled Results 2.6, it would be helpful if you could specify which genes you consider as the 'key driver genes'. This will provide a clearer understanding of the molecular pathways being targeted.

Response: Thank you for your valuable suggestions. The key driver genes include Insr, Irs1, Srebf2, PI3K, Akt, Glut4. We have added this information in the revised manuscript in red.

  1. Discrepancy in PI3K/AKT Signaling:

The abstract mentions that "the possible mechanism could be attributed to the Rk1+Rg5 regulated insulin resistance by modulating PI3K/Akt signal pathway." However, I did not find specific experimental evidence either in vitro or in vivo related to the PI3K/AKT signaling pathway. Given the importance of verification in network pharmacology, it would be beneficial if you could address this.

Response: Thank you for your valuable suggestions. Firstly, through network pharmacology analysis, we speculated the possible mechanism of Rk1+Rg5 improved T2DM by modulating PI3K/Akt signal pathway. Potential targets of Akt1 were confirmed by the molecular docking. Then, in order to further elucidate the potential mechanism of Rk1+Rg5 mediated T2DM, transcriptome analysis was performed in PA-induced L6 cells. KEGG and PPI analysis showed that Rk1+Rg5 mainly regulated the PI3K/Akt signaling pathway in PA-induced L6 cells. Finally, for more clarity expound the possible mechanism of Rk1+Rg5 regulated insulin resistance by modulating PI3K/Akt signaling pathway, the expression of Akt and p-Akt protein in PA induced L6 cells were further detected by western blotting. We have added the result of Akt and p-Akt protein expression by western blotting in the revised manuscript in the section of 2.6.

In conclusion, I believe that the study presents interesting findings on the potential mechanisms of Rk1 combined with Rg5 in the treatment of T2DM. Addressing the above queries and concerns will further strengthen the manuscript's contribution to the field.

Reviewer 2 Report

Liu et al. explored the potential targets and functional mechanisms of Rk1+Rg5 in the treatment of type II diabetes mellitus.  The compounds of Ginseng have been used as a Chinese medicinal herb for treating DM, cardiovascular, and inflammatory diseases. After investigating the efficacy of Rk1+Rg5 against T2DM, the authors found that these two major active components, Ginsenoside Rk1 and Rg5, offer promising results. The paper presents novel and interesting findings but requires minor revisions to be accepted for publication.

Comments

1.      Methods give antibody details for GLUT4 and GAPDH catalog number, dilution used, etc.

2.      Please check the statistics carefully for Figure 2 C; at the end of the experiment, the body 144 weight of db/db group showed a significant increase compared to the db/dm group, whereas 145 Rk1+Rg5, particularly high dosage Rk1+Rg5, had a greater effect on lowering body 146 weight gain (Figure 2C).

3.      The authors must review the statistics in Figure 2 for ambiguity.

4.      Doesn’t correspond to the figure number.

5.      “Meanwhile, compared with the db/dm group, the AUC was increased by 64.3% in the db/db group (Figure 3D). As expected, gavaged with glucose significantly increased blood glucose levels at 30 minutes, while gradually decreased during the period of 30-120 min and returned to normal levels after 120 min in the db/dm group, Rk1+Rg5-L, Rk1+Rg5-H and metformin. The AUCs values in mice treated with Rk1+Rg5-L, Rk1+Rg5-H and metformin, were de-177 creased by 13.14%, 20.89% and 11.9%, respectively” which figure the data corresponds it is very confusing.

6.      It is better not to combine the 3E and 3F should be written in separate sentences. It doesn’t make sense to the readers and is hard to follow.

The information presented in Figure 2 needs to be rewritten more clearly and concisely. Additionally, any spelling, grammar, and punctuation errors should be corrected to ensure accuracy and readability

Author Response

  1. Methods give antibody details for GLUT4 and GAPDH catalog number, dilution used, etc.

Response: Thank you for your valuable suggestions. Done as suggested. GLUT4 (66846-1-Ig), Akt (60203-2-Ig), p-Akt (80455-1-RR) and GAPDH (10494-1-AP) were purchased from Proteintech Co. Ltd. (CA, USA). GLUT4 (1:1000 dilution), Akt (1:5000 dilution), p-Akt (1:5000 dilution) and GAPDH (1:10000 dilution) were incubated overnight at 4 °C. We have added it in the revised manuscript in red.

  1. Please check the statistics carefully for Figure 2 C; at the end of the experiment, the body 144 weight of db/db group showed a significant increase compared to the db/dm group, whereas 145 Rk1+Rg5, particularly high dosage Rk1+Rg5, had a greater effect on lowering body 146 weight gain (Figure 2C).

Response: Thank you for your valuable suggestions. It is our carelessness. At the end of the experiment, the body weight of db/db group significant increase compared to the db/dm group, whereas Rk1+Rg5 inhibited body weight gain (Figure 2C). Apparently, Rk1+Rg5-H showed better efficacy on reducing body weight compared to Rk1+Rg5-L. We have amended the result of Figure 2 C in the revised manuscript in red.

  1. The authors must review the statistics in Figure 2 for ambiguity.

Response: Thank you for your valuable suggestions. Done as suggested. We have carefully reviewed the statistics in Figure 2.

  1. Doesn’t correspond to the figure number.

Response: Thank you for your valuable suggestions. It is our carelessness. We have made carefully check the manuscript and correspond to the figure.

  1. Meanwhile, compared with the db/dm group, the AUC was increased by 64.3% in the db/db group (Figure 3D). As expected, gavaged with glucose significantly increased blood glucose levels at 30 minutes, while gradually decreased during the period of 30-120 min and returned to normal levels after 120 min in the db/dm group, Rk1+Rg5-L, Rk1+Rg5-H and metformin. The AUCs values in mice treated with Rk1+Rg5-L, Rk1+Rg5-H and metformin, were de-177 creased by 13.14%, 20.89% and 11.9%, respectively” which figure the data corresponds it is very confusing.

Response: Thank you for your valuable suggestions. It is our carelessness. After oral glucose administration, the blood glucose level of each group increased rapidly, reaching individual peaks at 30 min, and then gradually declined, reflecting the processes of glucose metabolism and absorption in vivo. However, the response to glucose in the db/db group was weaker than that in the db/dm group, which was an apparent impairment of the glucose tolerance. Meanwhile, compared with the db/dm group, the AUC in the db/db group was significantly increased (P < 0.001, Figure. 3D). After Rk1+Rg5-L, Rk1+Rg5-H and metformin treatment, the AUC were significantly reduced by 13.14%, 20.37% and 11.91%, respectively (Figure 3D). We have added this information in the revised manuscript in red.

  1. It is better not to combine the 3E and 3F should be written in separate sentences. It doesn’t make sense to the readers and is hard to follow.

Response: Thank you for your valuable suggestions. The insulin tolerance tests (ITT) was used to detect the speed and ability of the body to remove glucose after injecting the insulin to further verify the effect of Rk1+Rg5 on insulin sensitivity. As shown in Figure 3E, the ITT results showed that extrinsic insulin was less effective in db/db mice than that in db/dm mice. Supplementation of Rk1+Rg5 exhibited improved insulin tolerance, the glucose-lowing effect of insulin in the Rk1+Rg5 group mice was greater than that in the db/db group, and the blood glucose levels were significantly lower in the Rk1+Rg5 group at 30, 60, 90, and 120 min after insulin injection (Figure. 3E). Similarly, the AUC was markedly reduced after Rk1+Rg5 treatment, compared with the db/db group (P < 0.01, Figure. 3F). These results indicated that Rk1+Rg5 could alleviate insulin resistance in T2DM mice. We have reviewed this information in the revised manuscript in red.

Reviewer 3 Report

September 20, 2023

Manuscript Number: ijms-2626220

Title: Identification of potential mechanisms of Rk1 combination Rg5 2

in the treatment of type II diabetes mellitus by integrating network pharmacology and experimental validation

Authors: Yao Liu, Jingjing Zhang, Chao An, Chen Liu Qiwen Zhang, Hao Ding, Saijian Ma, and Wenjiao Xue

This manuscript reports a study on the anti-T2DM (type II diabetes) effect of the combination of ginsenosides Rk1 and Rg5 (Rk1+Rg5) by network pharmacology, molecular docking, and experimental validation. Using a series of network analyses, the authors successfully extracted 10 core targets of Rk1+Rg5 from over 3,000 T2DM-related targets and identified them into signaling pathways likely regulated by Rk1+Rg5. They proposed the PI3K/Akt pathway as an important target and demonstrated a possible Rk1/Rg5 binding structures, which was experimentally validated both in vivo and in vitro. The study is well designed, the analyzes are well conducted, the data are well presented, and the manuscript is well written. I believe this work is worth publishing in Int. J. Mol. Sci. I have only minor comments to improve the manuscript.

1) The docking was done with Akt1, but in table 1, I find Rk1 or Rg5 show the large binding interaction energy toward GRB2, HRAS, PTPN11, and MAPK1 as well. Any explanation should be added for clarity.

2) It would be worth discussing how the presented results on mice relate with human.

3) It would be better to rewrite the description of the docking results (page 3, line 112-127). For example, “alkyl bond” in the following description sounds a little strange. “The carbon atom in Akt1 formed four alkyl bonds with LYS389 (3.83 Å),…”. This likely corresponds to the hydrophobic interaction between Rk1 and Lys389 of Akt1. I suggest the authors to follow the description in a document of software (seems using Discovery studio visualizer) or some other article (e.g. Chemistry Research Journal, 2020, 5(2):32-52).

4) Figure 1 should be enlarged for better visibility.

Author Response

Comments on the Quality of English Language

The information presented in Figure 2 needs to be rewritten more clearly and concisely. Additionally, any spelling, grammar, and punctuation errors should be corrected to ensure accuracy and readability

Response: Thank you for your valuable suggestions. Done as suggested. 

  • The docking was done with Akt1, but in table 1, I find Rk1 or Rg5 show the large binding interaction energy toward GRB2, HRAS, PTPN11, and MAPK1 as well. Any explanation should be added for clarity.

Response: Thank you for your valuable suggestions. The targets of Akt1, GRB2, HRAS, PTPN11 and MAPK1 were screened by the topological properties of the PPI network. Combined with GO and KEGG pathway enrichment analysis, the result showed that Akt took part in more signaling pathways of Rk1 and Rg5 in the treatment of T2DM compared to other proteins. Therefore, we chose Akt1 for analysis by molecular docking. We have added this information in the revised manuscript in red.

2) It would be worth discussing how the presented results on mice relate with human.

Response: Thank you for your valuable suggestions. Our study showed that Rk1+Rg5 significantly improved the hyperglycemic state of db/db mice, alleviated dyslipidemia, and promoted skeletal muscle glucose uptake. The underlying mechanism of Rk1+Rg5 may involve ameliorating insulin resistance and activating Akt to promote glucose transporter protein 4 (GLUT4) translocation to the cell membrane for glucose uptake. For people with T2DM, the primary target organs for absorption and consumption of postprandial glucose are the liver, adipose tissue, and skeletal muscle [1]. Skeletal muscle is responsible for 80% of these glucose absorption and consumption, making it an essential organ in regulating systemic glucose homeostasis and improving T2DM [2-3]. Activates the Akt signaling pathway, modulates the insulin resistance and induces the translocation of GLUT4 storage vesicles from the cytoplasm to cell membrane, which is essential to increases glucose uptake in skeletal muscle and improve T2DM [4]. Therefore, the pharmacological mechanism of treating T2DM is clear. Further validation is needed of experiments on humans. But we believe that Rk1+Rg5 may improve symptoms of T2DM on the human.

References

[1]. Mu, Y., MSCs improved insulin resistance and beta cell function in type 2 diabetes through modulation of macrophage polarisation and restoration of autophagy in insulin-targeted organs and islets. Diabetologia 2018, 61, S190-S191.

[2]. Merz, K. E.; Thurmond, D. C., Role of Skeletal Muscle in Insulin Resistance and Glucose Uptake. Compr. Physiol. 2020, 10 (3), 785-809.

[3]. Mendoza, C.;  Hanegan, C.;  Sperry, A.;  Vargas, L.;  Case, T.;  Bikman, B.; Mizrachi, D., Insulin receptor-inspired soluble insulin binder. Eur. J. Cell Biol. 2023, 102 (2), 8.

[4]. Richter, E. A., Is GLUT4 translocation the answer to exercise-stimulated muscle glucose uptake? Am. J. Physiol.-Endocrinol. Metab. 2021, 320 (2), E240-E243.

3) It would be better to rewrite the description of the docking results (page 3, line 112-127). For example, “alkyl bond” in the following description sounds a little strange. “The carbon atom in Akt1 formed four alkyl bonds with LYS389 (3.83 Å),…”. This likely corresponds to the hydrophobic interaction between Rk1 and Lys389 of Akt1. I suggest the authors to follow the description in a document of software (seems using Discovery studio visualizer) or some other article (e.g. Chemistry Research Journal, 2020, 5(2):32-52).

Response: Thank you for your valuable suggestions.Done as suggested. Based on the analysis of molecular docking (Figure 1F), Rk1 binds with Akt1 through alkyl hydrophobicity at positions of Pro318, Leu362, Met363, Lys386 and Lys389, van der Waals forces at positions of Tyr38, Arg41, Pro42, Asp44, Glu322, Asp387, Pro388, and hydrogen bonds at positions of Lys39, Glu40, Gln43, and Asp325. It can be seen from Figure 1G, Rg5 binds with Akt1 through alkyl hydrophobicity at positions of Tyr18, Cys310, Lys297, Van der Waals forces at positions of Leu155, Glu278, Thr312, Tyr315, Gly311, Leu295, Cys296, Arg273, Glu17, Ile84, Gly294, Lys276, Phe293, Met281, Tyr229, Phe236, hydrocarbon bonds at positions of Leu156, Asn279, Thr291 and hydrogen bonding at positions of Lys154, Asp274, Glu234. We have reviewed this information in the revised manuscript in red.

4) Figure 1 should be enlarged for better visibility.

Response: Thank you for your valuable suggestions. Done as suggested.
